# The Effects of Ethanol on the Heart: Alcoholic Cardiomyopathy

**DOI:** 10.3390/nu12020572

**Published:** 2020-02-22

**Authors:** Joaquim Fernández-Solà

**Affiliations:** 1Alcohol Unit, Internal Medicine Department, Hospital Clínic, Institut de Recerca August Pi i Sunyer (IDIBAPS), University of Barcelona, 08007 Catalunya, Spain; jfernand@clinic.cat; 2Fisiopatología de la Obesidad y la Nutrición, Instituto de Salud Carlos III, 28029 Madrid, Spain

**Keywords:** ethanol, alcohol, heart damage, alcoholic cardiomyopathy

## Abstract

Alcoholic-dilated Cardiomyopathy (ACM) is the most prevalent form of ethanol-induced heart damage. Ethanol induces ACM in a dose-dependent manner, independently of nutrition, vitamin, or electrolyte disturbances. It has synergistic effects with other heart risk factors. ACM produces a progressive reduction in myocardial contractility and heart chamber dilatation, leading to heart failure episodes and arrhythmias. Pathologically, ethanol induces myocytolysis, apoptosis, and necrosis of myocytes, with repair mechanisms causing hypertrophy and interstitial fibrosis. Myocyte ethanol targets include changes in membrane composition, receptors, ion channels, intracellular [Ca^2+^] transients, and structural proteins, and disrupt sarcomere contractility. Cardiac remodeling tries to compensate for this damage, establishing a balance between aggression and defense mechanisms. The final process of ACM is the result of dosage and individual predisposition. The ACM prognosis depends on the degree of persistent ethanol intake. Abstinence is the preferred goal, although controlled drinking may still improve cardiac function. New strategies are addressed to decrease myocyte hypertrophy and interstitial fibrosis and try to improve myocyte regeneration, minimizing ethanol-related cardiac damage. Growth factors and cardiomyokines are relevant molecules that may modify this process. Cardiac transplantation is the final measure in end-stage ACM but is limited to those subjects able to achieve abstinence.

## 1. Introduction

Ethyl alcohol, also known as “ethanol” or usually just as “alcohol”, is the most consumed drug in human history [1]. At present, its consumption rates are still very high, with a widespread worldwide distribution, in a global uncontrolled scenario with easy access [2]. In fact, there is an increasing consumption in particular groups, such as adolescents and young people [3,4]. 

This ethanol misuse at high consumption rates causes a variety of health problems, ethanol being the sixth most relevant factor of global burden of disease and responsible for 5.3% of all deaths [5]. Despite this clear epidemiological evidence of ethanol’s unsafe consumption and increased health risk, results of consumption policies are not effective enough. Therefore, the need to establish a more effective control on ethanol consumption has been repeatedly claimed [2]. 

It has been said that ethanol is the “perfect drug” because of its pleasant effects but damaging long-term effect [1,6]. It is distributed worldwide, with easy social access, and is pleasant when consumed, with positive sensations of welfare, but its negative effects, which include depressive and damaging noxious health effects, are reserved for later. This dual effect creates an additional difficulty to achieve an effective control. Ethanol is one of the most addictive drugs for humans, with high physical and psychological addiction potential [7]. Efforts to control alcohol addiction have just 50%–60% positive results in specific cessation programs [8,9].

One of the characteristics that makes ethanol harmful is its systemic toxic effect on the human body [10,11]. It has been described as having some kind of effect in all human body organs either in acute or chronic consumption [11,12]. The liver is the most affected organ, since ethanol is mostly metabolized there [11,13], but gastrointestinal, central, and peripheral nervous systems; the heart and vascular system; endocrinological systems; nutrition; and musculo-skeletal systems are clearly affected [10]. In addition, ethanol is an immunosuppressive drug that is pro-inflammatory and pro-oncogenic [14,15,16,17]. 

In fact, the particular effects that ethanol produces in a specific organ depend on several factors [18,19]. One is the physical characteristics of ethanol itself, with a low molecular size, high distribution capacity, and high tissue reactivity. Ethanol has active metabolites (acetaldehyde-acetate, fatty-acid ethyl esters), is a potent enzymatic inductor [20,21], and interacts with other drugs [22], but its effect also depends on the characteristics of the target organ, with excitable (brain, heart) [23] and metabolic (liver, pancreas) organs [24] being more susceptible to ethanol-induced organ damage. In addition, there is a relevant role on each organ, particularly on defense and adaptive mechanisms, with a clear induction of anti-oxidant, metabolic, and anti-inflammatory protective responses as a result of ethanol aggression [18,25,26]. This multi-factorial effect is attributed to genetic factors [27] and ethnic [28] variability. The final damage is an equilibrium between the intensity of damaging effects and the possibility of defense, plasticity, regeneration, and adaptation for every specific organ [29,30,31]. Thus, alcohol-dilated cardiomyopathy (ACM) is the result of dosage and individual predisposition [32].

The cardiovascular system is, after the liver and gastrointestinal system, the second most affected system by global ethanol toxicity [1,33,34]. At a high dose (more than 60 g/day for men and 40 g/day for women) and chronic consumption (usually more than 10 years), ethanol increases the atherosclerosis process with coronary, cerebral, and peripheral vascular involvement [35,36], increases arterial hypertension [37], and causes progressive myocardial damage (alcoholic dilated cardiomyopathy—ACM) [18,38,39,40,41] as well as induction of arrhythmias [42,43]. The effect of a low dose of alcohol consumption on the cardiovascular system has been also extensively evaluated with evidence of a dual effect, beneficial for coronary artery disease at low doses [44] but reversing to a damaging effect at moderate to high doses [19]. Although there is beneficial potential in some patients, the coexistence of increased risk of cancer, neurological brain damage, and the high risk of ethanol addiction makes it necessary to discourage this low-dose consumption in the general population [19,41,45]. Specific caution should be recommended regarding children or adolescents [4] and women [46], who are more susceptible to the damaging effects of ethanol at the same doses of consumption as men. Similarly, patients suffering from other ethanol-related diseases such as liver cirrhosis or brain atrophy should completely suppress their ethanol consumption [47,48]. Therefore, the only safe ethanol dose for the cardiovascular system is zero [41,45,49,50,51]. 

In this review, we specifically describe and discuss the global effects that ethanol exerts on the heart myocytes, the so-called alcoholic cardiomyopathy (ACM).

## 2. What is Alcoholic Cardiomyopathy

The first clinical recognition of ACM was performed by Hippocrates in Greece during the 4th century B.C. However, its modern clinical report was delayed until the 19th century, where specific ACM cases were clinically described in Germany and England [1]. During the 20th century, the physiopathological basis for ACM was progressively established [6]. At present, ACM is defined as a dilated cardiomyopathy of toxic origin with low left-ventricle ejection fraction, chamber dilatation, and progression to congestive heart failure [18,52,53]. Excessive EtOH consumption is one of the main causes of non-ischemic dilated cardiomyopathy (CMP), representing around one-third of cases [30].

### 2.1. The Natural Course of ACM

At clinical level, the course of ACM is similar to idiopathic dilated CMP [54]. The subject with excessive alcohol consumption, after more than 10 years of high ethanol consumption, usually develops subclinical heart functional changes before symptom appearance or signs of heart failure [55,56]. These may be detected with echosonography in around one-third of high-dose chronic consumers with preliminary evidence of subclinical left-ventricle (LV) diastolic dysfunction before progression to subclinical LV systolic dysfunction [57].

Symptoms of ACM are not specific and overlap with other forms of heart failure [30,41,58]. They appear when ventricle dilatation, hypertrophy, and dysfunction are established. Later and progressively in the course of the disease, around 20% of women and 25% of men with excessive alcohol consumption develop exertion dyspnea and orthopnea, leading to episodes of left-ventricle heart failure [39,46,59]. Depression of LV ejection fraction (EF) is the hallmark of this period that also occurs with a reduction in LV shortening fraction, increase in LV diameter, and mass indices that may be measured by echocardiography or cardiac MR spectroscopy [40,52]. Congestive symptoms, such as the expression of right ventricular failure, with peripheral edema or anasarca, are characteristic of advanced cases of ACM [42,56]. 

A diverse variety of arrhythmias appear early and may worsen the course of ACM, atrial fibrillation being the most frequent [60] and ventricular tachycardia the most deleterious [61]. These arrhythmias are usually related to episodes of binge drinking [43,62] and are more frequent in established ACM than in subjects with normal cardiac function [52]. In chronic alcoholics, arrhythmia may frequently appear in relation to episodes of ethanol abstinence because of the increased release of catecholamines and electrolyte deficiencies [19].

Along with developing heart damage, patients with ACM may also damage other organs, such as the liver, central and peripheral nervous system, skeletal muscle, pancreas, and digestive tract, and are exposed to an increased risk of cancer [24,63,64]. In fact, ACM is related to systemic damage induced by ethanol misuse and its global biological response [10,11,31]. 

Mortality in ACM is related to the progression of heart failure and malignant arrhythmias [58,65]. In long-term follow-up studies, a mortality rate of 10% of patients/year has been observed in the group of patients with persistent high-dose ethanol consumption [19,52].

### 2.2. Is ethanol the Real Cause of ACM

Before recognizing that ethanol itself is the etiological factor of ACM, different theories and hypotheses emerged [1,66]. It was suspected that malnutrition, frequently related to chronic alcohol misuse, was the origin of ACM [6,67]. However, it has been evidenced that ACM may develop in the absence of protein or caloric malnutrition [38]. However, nutritional factors may worsen the natural course of ACM and should be avoided [18,19]. 

Occidental Berberi is the term used for the clinical scenario caused by thiamine deficit, a situation commonly present in chronic alcohol misuse, and was attributed as the cause of ACM [68,69]. Similarly, electrolyte (Na, K, Ca, Mg, P) deficiencies or disturbances may play a major role in cardiac function, and ethanol misuse may be related to them [52]. Selenium deficit (Keshan disease in China) could also induce ACM in specific areas [70].

Another curious hypothesis from Germany suspected that some ethanol additives, such as anti-foam beer products with arsenic or cobalt content, produced cardiac toxicity and development of ACM [71]. However, there is no clear demonstration of this contaminant effect. Therefore, it is evident that ACM may develop with normal serum thiamine and electrolyte levels [38,66]. Consumption of other drugs such as cocaine or tobacco may interact with ethanol and potentiate the final ethanol-related cardiac damage [22,72].

### 2.3. Ethanol or Acetaldehyde

One relevant question concerning ethanol cardiac toxicity is if ethanol itself or its active metabolite acetaldehyde causes cardiac damage [73,74]. In fact, both molecules are directly cardiotoxic, decreasing structural protein synthesis and heart contractility and increasing oxidative and metabolic damage, leading to autophagy [20,75]. In experimental studies, acetaldehyde directly impairs cardiac contractile function [76], disrupts cardiac excitation–contraction coupling, and promotes oxidative damage and lipid peroxidation [20]. Acetaldehyde is produced at a lower quantity in the heart as compared to the liver, and systemic acetaldehyde does not achieve toxic heart concentrations [77]. In addition, acetaldehyde is able to interact with proteins and produce protein-adduct compounds that are highly reactive and may induce additional inflammatory and immunologic heart damage [78]. Therefore, because of its multiple actions, acetaldehyde may influence ACM pathogenesis in addition to ethanol effect itself [20,76,77]. 

### 2.4. The dose-Related Effect of Ethanol and Beverage Types on the Heart

Until the second part of the 20th century, there was no scientific evidence on the direct and dose-dependent effect of ethanol on the heart as cause of ACM [6,38]. This is a longstanding accumulated effect that usually appears when a subject has, in their lifetime, consumed more than 7 Kg of ethanol per Kg of body weight in men (equivalent to 60 drinks per month), and 5 Kg of ethanol per Kg of body weight in women (equivalent to 43 drinks per month) [19,46]. However, there is a clear personal susceptibility of this effect that creates a wide variability range and supposes significant inter-individual differences [50,66]. In fact, ACM is considered to be the result of dosage and individual predisposition [32]. 

Concerning the different effects of beverage choice, ACM may develop through the consumption of any type of beverage, such as wine, beer, or spirits, in a lineal dose-dependence relationship with the total lifetime dose of ethanol consumed by an individual [38]. In general, alcoholic patients consuming >90 g of alcohol a day (approximately seven to eight standard drinks per day, considering a standard drink 12–15 g of alcohol) for >5 years are at risk for the development of asymptomatic ACM [18]. Wine is considered less damaging compared to other alcohol beverages, probably because of its antioxidant polyphenolic content, with molecules such as resveratrol [79,80]. The consumption of spirits that contain greater ethanol content may easily induce binge drinking and higher cumulated lifetime dose of ethanol, increasing the risk of ACM [19].

### 2.5. The effects of Moderate Consumption of Ethanol and Binge-drinking 

Moderate drinking, considered as the consumption of 20–60 g/day in men (1.5–4 standard drinks) and 10–40 g/day in women (1–3 standard drinks), usually is not associated with significant cardiotoxicity [19,44]. It does not suppose a risk of ACM development unless consumed over a large period of time (more than 10 years) [19,52]. Moderate alcohol consumption has been associated with lower risk of heart failure in prior studies of healthy individuals [52] and appears equivalent to abstention in improving LV ejection fraction among heavy drinkers with established ACM [81,82]. 

Binge drinking, defined as the consumption in men of five or more drinks and four or more drinks in women in about two hours, is clearly detrimental for the heart [83,84]. It brings a person’s blood ethanol concentration (BAC) to 0.08 grams or higher [85]. It causes acute myocardial effects with a temporary depression of LV EF evident in experimental [85,86] and clinical models [87,88]. Is more frequent in subjects with LV EF < 40% than in those with preserved LV EF [56]. Acute ethanol binge drinking also induces a variety of arrhythmias, known as “Holiday heart Syndrome” [43]. All these acute effects produce impairments on the natural course of chronic ACM [62]. Spirits and other beverages containing a high percentage of alcohol are more detrimental than wine consumption regarding the induction of acute cardiac effects [31,80].

### 2.6. The Effect of Low-dose Ethanol on ACM

Low-dose ethanol consumption, considered as the daily consumption of up to one standard drink for women and two standard drinks for men, has a beneficial effect on preventing coronary heart disease [44], heart failure [87], and global mortality [89] as assessed in multiple clinical and epidemiological studies with a clear “J-shape” curve of effect [58,90]. However, it is possible with low-dose alcohol consumption to achieve an accumulated lifetime dose of ethanol reaching the threshold level required to develop ACM in long-term susceptible consumers [19]. This is especially possible in those patients more sensitive to the toxic effects of ethanol on the heart, such as women [46] and patients with other systemic diseases related to ethanol (cirrhosis, malnutrition, or neurological damage). In addition, some genetic polymorphisms, such as the “*DD*” isoform for the angiotensin-converting enzyme gene [91] and titin truncated-variants [92], are associated with higher genetic vulnerability to ACM. Therefore, there is no safe dose of ethanol consumption to completely avoid the development of alcohol CMP, with complete abstinence being recommended in susceptible subjects [41,45,51]. 

### 2.7. Gender Differences in ACM

One of the relevant facts in ACM is the existence of a clear gender difference, women being more susceptible to the toxic effects of alcohol than men at the same level of lifetime ethanol consumption [93,94]. This fact has been assessed with echocardiographic monitoring in women consuming high doses of ethanol both in the subclinical period of disease [46] as well as in the clinical period when congestive heart failure appears [95]. At the experimental level, some gender differences also are evident in functional proteomic analysis, with sex-dependent differences in structural and energy-producing myocardial proteins in a rat model of alcoholic cardiomyopathy [96]. The biological reason for this gender difference is based on different ethanol absorption rates, distribution pattern, and metabolism in women compared to men [52]. Therefore, efforts to prevent ACM development in women should be specifically addressed [97]. During pregnancy, ethanol consumption should be clearly discouraged because of the possibility of fetal alcohol syndrome or the development of other congenital heart diseases [97].

## 3. Pathological Aspects of ACM

In the course of ethanol-induced cardiac damage, one of the more relevant findings is that ethanol exerts its deleterious effects on cardiac myocytes at multiples sites (membrane, receptors, mitochondria, ribosomes, sarcolemma, DNA, or cytoskeleton) [18,19,98] (Table 1). 

This is because the ethanol molecule has a small size and is highly reactive, with many cell targets. In addition, ethanol has a widespread diffusion because of the potential for distribution though biological membranes, achieving targets not only in the membrane receptors and channels but also in endocellular particles and at the same nuclear compartment [29,99,100]. This induces a variety of effects, since more than 14 different sites in the myocyte can be affected by ethanol [19,98]. Thus, ethanol enhances permeation in model membranes by interfering with plasma membrane composition and permeability [99], disturbing signaling mechanisms, and activating apoptosis [101], as well as disturbing L-Type Ca^2+^ channel activity [85,86], Na^+^/K^+^ ATPase channel activity [102], Na^+^/Ca^2+^ exchanger activity, and Na^+^ and K^+^ channel currents [19,29]. Specifically, ethanol disturbs the ryanodine Ca^2+^ release, the sarcomere Ca^2+^sensitivity [102,103], the excitation–contraction coupling and myofibrillary structure, and protein expression, decreasing heart contraction [86]. Ethanol-induced disruption of ribosomal protein synthesis also contributes to non-contractile protein depletion [104]. Several aspects of mitochondrial function, including respiratory complex activities and mitochondrial-dependent oxidative damage and apoptosis, are also induced by ethanol [26,100]. Myocyte cytoskeletal structure [21], connexin channel communication, and desmosomal contacts are affected by ethanol, causing structural cell instability [105]. Ethanol may induce changes in nuclear regulation of transcription with a dose-dependent translocation of NFkB into the nucleus [106]. The resulting effect in those multiple sites may be additive and synergistic, increasing the final damage [20,52] (Figure 1).

### 3.1. Oxidative and Energy Disturbances in ACM

Since myocardium requires a high energy supply to maintain persistent sarcomere contractions, it was supposed that alcohol could exert its damaging effect on the mitochondrial energy supply system, with the disruption of oxidative control mechanisms [26,100]. In fact, mitochondrial structural changes have been described in chronic alcohol consumers, with swollen megamitochondria and the distortion of inner cristae [107,108]. Functionally high ethanol produces disruptions in the myocyte oxidative pattern and decreases in Complex I, II, and IV of the mitochondrial respiratory chain [100,109,110]. As a reflection of this metabolic derangement, cytoplasmic lipid droplets and glycogen deposits appear.

At ultrastructural level, dysfunction on the transition pore in the inner membrane is related to ethanol exposure [111]. In addition, ethanol induces mitochondrial-dependent apoptosis pathways with Bax and caspase activation [101]. 

### 3.2. Ethanol-induced Myocyte Apoptosis and Autophagy

Myocyte apoptosis, based on assessment of TUNEL staining and caspase activity, has been demonstrated to be an active phenomenon leading to myocyte loss in diverse cardiomyopathies [113,114] and also in chronic high-dose ethanol consumption both in experimental [109] and clinical models [101]. Apoptosis may be induced by ethanol through mitochondrial membrane permeabilization and the release of pro-apoptotic factors (cytochrome c) from the mitochondrial inter-membrane space to the cytosol. Chronic ethanol exposure, in combination with other stress signals, provides a trigger for cardiac apoptosis through activation of the mitochondrial permeability transition pore by physiological calcium oscillations [111]. 

However, cardiac apoptosis may also develop independently of the mitochondrial pathway [115] through the extrinsic pathway, which involves cell surface death receptors [116]. In addition to inducing apoptosis, ethanol inhibits the effect of anti-apoptotic molecules such as BCL-2 [101]. Ethanol-induced myocyte apoptosis may be regulated by growth factors [117,118] and cardiomyokines [119]. The percentage of apoptotic myocytes in ACM is relatively low but, in combination with a persistent decrease in myocyte proliferation, they may contribute to an absolute cell loss and decreased cardiac contractility [52,115]. Recent data favored a role for micro RNA, such as the involvement of miR-378a-5p in cardiomyocyte apoptosis and ACM development through ALDH2 gene suppression [120].

Recently, apoptosis and necrosis have been also attributed to autophagy in ACM [18]. In order to maintain cardiac homeostasis, the removal of defective organelles and cell debris by autophagy is essential both in physiological and pathological conditions [115]. Dysregulated excessive autophagy, together with other factors such as oxidative stress, neurohormonal activation, and altered fatty acid metabolism, contributes to cardiac structural and functional damage following alcoholism. This influences the maintenance of cardiac geometry and contractile function, increasing the development of ACM [121]. In ACM, protein degradation with sarcomere disarray and contractile protein loss has been suggested to be a key point of autophagy induction [18]. Different pathogenic hypotheses have been suggested, such as the pivotal role of acetaldehyde [122], the role of oxidative stress and stress signaling cascades [109], and the translocation of NFkB into the nucleus [106]. Although the mechanism of action behind autophagy and its signaling regulatory cascades remains elusive in ACM [121], its understanding may contribute to better identifying molecular mechanisms underlying the early stages of alcoholic cardiomyopathy and suggest novel strategies to counteract the integrated risk of cardiotoxicity in chronic alcohol consumption [106].

### 3.3. Ethanol-induced Heart Fibrosis 

After myocyte apoptosis or necrosis, the heart tries to repair and regenerate this tissue damage [39,123], but the heart regenerative capacity is low as a result of the ethanol aggressive damage and develops ineffective repair mechanisms such as progressive fibrosis [124,125]. In fact, ethanol itself decreases the myocyte regeneration capacity and increases the fibrogenic process [52,126]. Subendocardial and interstitial fibrosis progressively appear in the course of ACM, usually in advanced stages [52,56]. More than 30% of the myocyte ventricular fraction can be replaced by fibrotic tissue, thus decreasing the heart elasticity and contractile capacity [64] (Figure 2). Some cardiomyokines, such as FGF21, may regulate this process of alcohol-induced cardiac fibrosis [119].

### 3.4. Ethanol Disruption of [Ca^2+^] Transients and SR Activation

Since cardiac myocytes are excitable cells, and ethanol may easily damage this excitation–contraction mechanism, disruption of this coupling mechanism is involved in the ACM pathogenic process [19,58]. Ethanol may produce the modification of sarcolemmal membrane L-type Ca^2+^ channels, leading to a decrease in transmembrane electrically induced Ca^2+^ transients [85,103,127]. One of the most relevant targets of ethanol in the membrane is the disruption of membrane receptor composition and activities [86]. The ryanodine L-type Ca^2+^ receptor at the sarcoplasmic reticulum (SR) is also significantly affected by ethanol in a dose-dependent manner [86,102]. This causes a decrease in sarcolemmal contraction and also disturbance in other intercellular organelles dependent of i.c. [Ca^2+^] transients [102]. As an adaptive process, chronic alcohol consumption induces up-regulation of myocardial L-type [Ca ^2+^] channel receptors, whose activity decreases in the presence of cardiomyopathy [103]. 

### 3.5. Sarcomere Damage and Dysfunction in ACM 

Chronic ethanol misuse clearly depresses protein synthesis and degradation, involving both structural and non-structural heart proteins [104,128]. At a pathological level, sarcomere Z-line distortion and disruption of the sarcomere pattern leads to myocytolysis [107,129]. Myocytolysis is evident through focal myofiber dissolution, cell vacuolization, and fiber disarray [19] (Figure 2). The sarcomere complex is early affected by ethanol, decreasing the titin content, a protein that is responsible for sarcomere relaxation and LV distensibility [130]. Ethanol also decreases myofilament Ca^2+^ sensitivity [20]. This damage first induces diastolic dysfunction, which is initially subclinical and later clinically apparent [57]. In addition, contractile sarcomere proteins such as Myosin, Actin, and Troponin are also affected by ethanol, causing the functional progressive depression of myocyte contractility, inducing progression to heart failure [56,104,131].

### 3.6. Cardiac Hypertrophy and Remodeling in ACM

Cardiac remodeling is a global process that myocardium establishes as a result of different aggressions [31,132]. Heart myocytes are relatively resistant to the toxic effect of ethanol, developing a functional and structural compensatory mechanism able to minimize or repair the ethanol-induced myocyte damage [20,31,39]. Structurally, hypertrophy of myocytes is seen in the early stages to avoid contractile depression [52,107,125]. Myocytolysis progressively develops, disturbing the sarcomere contractile system. The ventricles show wall hypertrophy and compensatory dilatation. The heart output is progressively lower in a dose-dependent relationship with the lifetime accumulated total dose of alcohol consumed [38]. Several growth factors and cardiomyokines exert an autocrine or paracrine effect that tries to compensate for this heart damage [119,133]. Antioxidant, anti-inflammatory, anti-apoptotic, and antifibrogenic mechanisms try to avoid myocyte necrosis and heart fibrosis [14,30,58]. The final result is that achieved from the equilibrium between the degree of damage and the capacity of heart repair mechanisms in each specific individual [31,56]. 

### 3.7. End-stage ACM

The heart repair mechanisms that minimize the ethanol-induced cardiac damage are limited and apparently ineffective in chronic longstanding scenarios [56,124]. This is usually after more than 20 years of high ethanol consumption at cumulated lifetime doses higher than 20 Kg ethanol/Kg body weight, equivalent to 180 drinks per month [52,134]. The histological pattern of this situation is a diffuse myocyte necrosis that is being substituted by interstitial fibrosis and compensatory fiber and nuclei hypertrophy of the remaining myocytes [64]. In this scenario, the subject with LV ejection fraction < 15% develops frequent episodes of congestive heart failure and ventricular arrhythmias [43,61,131]. Systemic involvement of ethanol is usually present, with coexistent liver cirrhosis and neurological damage, a fact that worsens the patient prognosis. The mortality of this situation is higher than 30% per year, mainly affecting those subjects who persist in ethanol consumption [52,54,134]. 

## 4. Prognosis of ACM

The natural course of ACM is mainly related to the degree of persistence in alcohol consumption and the individual biological adaptive response [2,20,41,56,81]. Ethanol abstinence allows for recovery in the majority of cases, including in those with previous severe depression of LV EF [81,88,135]. On the contrary, subjects who continue drinking at moderate to high doses (more than 60 g ethanol/day in men—equivalent to four standard drinks—and 40 g of ethanol/day in women—equivalent to 2.5 standard drinks—experience progressive functional and structural cardiac impairment, with repeated episodes of cardiac left or congestive failure, arrhythmias, and progression to death, with a mortality rate of 10%/year [55,61]. Episodes of binge-drinking are highly damaging and should be especially avoided [83]. In these subjects, mortality is related to episodes of sudden death and refractory congestive heart failure [42,131,136]. In addition to the risk of other ethanol-mediated systemic diseases, liver cirrhosis is the main risk that is highly prevalent in ACM [63]. In a follow-up study on ACM, the independent predictors of all-cause mortality were the QRS duration, systolic blood pressure, and New York Heart Association classification [137]. In another long-term outcome of alcoholic and idiopathic dilated cardiomyopathy, multivariate analysis in the entire cohort demonstrated that increased pulmonary capillary wedge pressure, alcoholism, and lack of abstinence during follow-up and decreased standard deviation of all normal-to-normal R–R intervals were independent predictors of cardiac death [54].

Therefore, complete abstinence from ethanol is the most useful measure to control the natural course of ACM [51,56,135]. In fact, patients with ACM who abstain from alcohol have a better long-term prognosis than subjects with idiopathic dilated CMP [54]. Out of end-stage cases, the majority of subjects affected by ACM who achieve complete ethanol abstinence functionally improve [33,82,135]. However, subjects with ACM are mostly alcohol-dependent [66,138]. The percentage of effective abstinence achievement on these patients submitted to specific programs ranges from 50% to 60% [8,9]. Therefore, many ACM subjects are not able to effectively control their alcohol-consumption rates. In this group of subjects, it has been demonstrated that the strategy of controlled drinking, that supposes to decrease the degree of ethanol consumption to less than 60 g/ethanol/day is still effective, achieving significantly functional cardiac improvement, although to a lesser extent when compared to that obtained in total abstainers [82]. Therefore, any decrease in the previous quantity of alcohol consumption may improve, to some degree, cardiac health [51]. Since ACM is related to frequent perioperative events and high postoperative morbidity [139], detection and treatment of ACM is compulsory to avoid anesthetic and surgical complications [140]. 

## 5. Treatment of ACM

The treatment of episodes of heart failure in ACM does not differ from that performed in idiopathic-dilated CMP [52,54]. A decrease in cardiac preload with diuretics and postload with angiotensin-converting-enzyme inhibitors or beta blockage agents allows for an improvement in signs of acute heart failure [19,131]. A reduction of dietary sodium intake is also necessary. Nutritional factors are relevant in ACM [67]. A Mediterranean diet, based on monounsaturated fats from olive oil, fruits, vegetables, whole grains, and legumes/nuts, has been demonstrated to be beneficial for primary prevention of global cardiovascular events (myocardial infarction, stroke, or death from cardiovascular causes) [80,141,142]. However, since it includes moderate alcohol consumption of red wine, this aspect should be clearly avoided in subjects affected by ACM. The exact mechanism by which an increased adherence to the traditional Mediterranean diet exerts its favorable effects is not known. However, its beneficial cardiovascular effect may be caused by different factors including lipid-lowering, protection against oxidative stress, inflammation and platelet aggregation, modification of hormones and growth factors, inhibition of nutrient-sensing pathways by specific amino acid restriction, and gut-microbiota-mediated production of metabolites influencing metabolic health [143]. 

In ACM, it is relevant to consider the treatment of the other alcohol-induced systemic damage, such as liver cirrhosis, malnutrition, and vitamin and electrolyte disturbances [2,11,52]. Notably, in patients with a history of chronic alcohol consumption complicated by significant myocardial dysfunction and chronic malnutrition, re-feeding syndrome may increase the cardiac dysfunction. Therefore, physicians should be aware of the risk of new cardiomyopathy in patients with these overlapping diagnoses [144]. Control of these alcohol-related systemic diseases, as well as the strict control of the presence of other heart risk factors (tobacco, cocaine, arterial hypertension, diabetes mellitus, or anemia) contributes to ACM improvement [10,20,23,37,52]. Atrial fibrillation should be controlled with chronotropic drugs such as digoxin or diltiazem and anticoagulant treatment to avoid arterial embolisms [60,145].

In end-stage disease (LV EF < 15%), cardiac transplantation may be the only possibility [146]. In those subjects, systemic alcohol-related damage (cancer, liver cirrhosis, or dementia) should also be excluded. However, most programs of alcohol transplantation require the guarantee of a long period of ethanol abstinence (almost 3 months), a condition not accomplished by most subjects with ACM, who are alcohol-dependent [9,66]. Therefore, few patients with end-stage ACM receive heart transplantation. In a series of 94 chronic alcoholics with ACM, only 15% achieved heart transplantation [56].

New strategies to improve the natural course of ACM have been proposed as promising agents in this field [112,147]. Since ethanol has multiple cell targets with different pathological mechanisms implicated, those different strategies to directly target alcohol-induced heart damage are only partially effective and can only be used as support medication in a multidisciplinary approach [112]. They try to control myocardial remodeling to avoid the progression of myocyte hypertrophy [39,148] or fibrosis [149] and ventricle dysfunction and dilatation, as well as to increase the degree of myocyte regeneration [150]. Recently, new cardiomyokines (FGF21, Metrnl) and several growth factors (myostatin, IGF-1, leptin, ghrelin, miRNA, and ROCKs inhibitors) have been described as being able to regulate cardiac plasticity and decrease cardiac damage, improving cardiac repair mechanisms [112,119]. They aim to control oxidative damage, myocyte hypertrophy, interstitial fibrosis, and persistent apoptosis. Pharmacological restoration of autophagic reflux by inhibition of soluble epoxide hydrolase has been described to ameliorate chronic ethanol-induced cardiac fibrosis in an in vivo swine model [151]. In addition to these, stem-cell therapy tries to improve myocyte regeneration [112,152]. However, these new strategies have not yet demonstrated their real effectiveness in clinical trials, require further evaluation, and are not approved for clinical use [147]. 

## 6. Discussion and Conclusions 

Since ethanol consumption of the global population is not currently under control [2], the incidence of alcoholic cardiomyopathy is expected to be maintained in the future, especially in specific population groups, such as adolescents and young people [3]. Therefore, efforts for the prevention, early detection, and specific treatment in this relevant disease should be established [45]. The direct dose-dependent effect between alcohol intake and development of ACM is clearly established [50,52], women being more sensitive than men to the toxic effects of ethanol on the heart [46]. However, genetic polymorphisms, the use of other concomitant drugs (tobacco, cocaine), and the presence of other cardiac risk factors (hypertension, diabetes) may influence and worsen the natural course of ACM in each specific individual [27,72,98]. The multiple sites of myocyte damage from alcohol [11,19,23] and the genetically mediated individual predisposition [32,153] create a large individual clinical variability and make it difficult to establish a simple effective treatment for ACM [27,30,52]. Heart remodeling is an adaptive mechanism, susceptible to being modified in ACM by the use of cardiomyokines (FGF21, Metrnl) and growth factors (IGF-1, Myostatin) [112,119].

Since ethanol is a drug with systemic toxic effects, the evaluation of global alcohol-related systemic damage is necessary in ACM [2]. Control of other cardiac risk factors also allows for a better prognosis in ACM [72]. Total abstention from alcohol is the preferred goal [41,51], although controlled drinking (with daily consumption < 60 g/day) still allows improvement [82,135]. Binge drinking should be absolutely discouraged in ACM [83]. Subjects with ACM who continue in high-dose ethanol consumption have a bad prognosis, with repeated episodes of heart failure and ventricular arrhythmias leading to a 10% increase in annual mortality rate [56,61]. New strategies aiming to control apoptosis, autophagy and pathological heart remodeling, and increase myocyte regeneration may be promising in the near future [112,133]. However, areas of uncertainty in this complex disease are still present and should be further explored [30].

## Figures and Tables

**Figure 1 nutrients-12-00572-f001:**
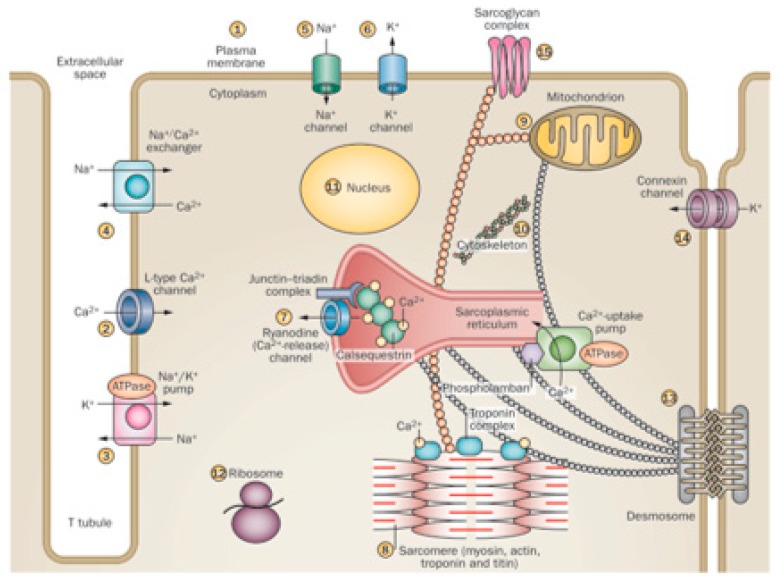
Different effects of ethanol on cardiomyocyte organelles. (Adapted form Nature 451; 929–936, 2008). Cardiac myocytes are excitable cells with complex signaling and contractile structures and are highly sensitive to the toxic effect of alcohol on: (1) plasma membrane composition and permeability, signaling, and activation of apoptosis; (2) L-Type Ca^2+^ channel activity; (3) Na^+^/K^+^ ATPase channel activity; (4) Na^+^/Ca^2+^ exchanger activity; (5) Na^+^ channel currents; (6) K^+^ channel currents; (7) ryanodine Ca^2+^ release; (8) sarcomere Ca^2+^ sensitivity, excitation–contraction coupling, myofibrillary structure, and protein expression; (9) several aspects of mitochondrial function, including respiratory complex activities; (10) cytoskeletal structure; (11) nuclear regulation of transcription; (12) ribosomal protein synthesis; (13) desmosomal contacts; (14) connexin channel communication; (15) sarcoglycan complex interactions.

**Figure 2 nutrients-12-00572-f002:**
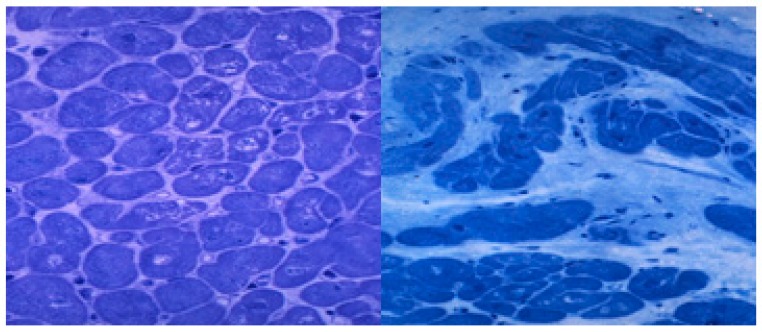
Histological lesions in the subclinical and clinical periods of human alcoholic cardiomyopathy. Left ventricle apical biopsy. Semithin section. Toluidine blue staining x 400 magnification. Left: Subclinical period with slight signs of myocytolysis, disarray, and myocyte hypertrophy (arrows). Right: End-stage clinical period with myocyte loss, diffuse interstitial fibrosis and intense nuclear and myocyte hypertrophy (white arrows).

**Table 1 nutrients-12-00572-t001:** Mechanisms of alcohol-induced heart damage and their effectors.

Mechanisms	Effectors
Interference with cell signaling and calcium transients	MAPK, TGF-β, PKC, PPARγ, MMPs, NF-κβ, PAI-1
Decrease in excitation–contraction coupling mechanisms	intracellular [Ca]^2+^ transients, L-type Ca^2+^ channel
Induction of oxidative damage	ROS, SOD, acetaldehyde
Pro-inflammatory effect	IL-2, TNF-α, NF-κβ
Induction of apoptosis	FAS, TNF-α, TGF-β, Bax-Bcl-2, caspases 3,6
Induction of fibrosis	TLR-4, TGF-β
**Protein-adduct formation**	protein–ethanol adducts
	malondialdehyde–DNA adducts
Disruption in protein synthesis	decrease in ribosomal protein synthesis, actin, myosin, troponin, titin
Increased glycogen deposition	glycogen synthase kinase-3β, PARP
Renin–angiotensin–aldosterone activation	renin, angiotensin, aldosterone, p38 MAPK/Smad
Interference in hormone-growth factors	myostatin, ghrelin, leptin, IGF-1
Interference in regulatory cardiomyokines	FGF21
Decrease in myocyte regeneration	myostatin, IGF-1
Impairment of extracellular matrix turnover	cytoskeletal structure, connexin channel, desmosome contacts
Imbalance between cardiac lesions/repair mechanisms	cell apoptosis and necrosis increased myocardial fibrosis and decreased myocyte regeneration

Adapted from Fernández-Solà J and Planavila A, Int J Med Sci 2016, 17, 10. [112].

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
