# Peer review of "The Effects of Ethanol on the Heart: Alcoholic Cardiomyopathy"

_nutrients, 2020, doi:10.3390/nu12020572_

Round 1

Reviewer 1 Report

The authors reviewed ACM. It is scientifically interesting.

At first, there are too many spelling mistakes. Even EtOH is misspelled.

wiht in L.133, beberage in L.154, whine in L.155, and so on.

In Figure 1, I could not find no. 11. Figure 1 needs Figure legends.

Is Figure 1 needed? If so, describe about the relationships between EtOH and these molecules in the manuscript.

There are some unnecessary spaces.

Author Response

We thank this reviewer comments and suggestions.

Point-By-Point response:

1.- We agree that many spelling mistakes remain in the previous text version. Sorry, this is because we have performed the English correction in the final version of the manuscript.

2.- We have corrected those 2 mentioned  mistakes

3.- Nº 11 in Figure 1 is inside the circle that represents the nucleus (in yellow at the center).

4.- Figure 1 describes schematically different ethanol targets on the cardiac myocyte.  We consider this figure necessary. As suggested, we added explanation between the relationship of ethanol and these molecules in  figure 1 legend:

Fig 1 Legend: Cardiac myocytes are excitable cells with complex signaling and contractile structures and are highly sensitive to the toxic effect of alcohol on: (1) plasma membrane composition and permeability, signaling, and activation of apoptosis; (2) L-Type Ca2+ channel activity; (3) Na+/K+ ATPase channel activity; (4) Na+/Ca2+ exchanger activity; (5) Na+ channel currents; (6) K+ channel currents; (7) ryanodine Ca2+ release; (8) sarcomere Ca2+ sensitivity, excitation–contraction coupling, myofibrillary structure, and protein expression; (9) several aspects of mitochondrial function, including respiratory complex activities; (10) cytoskeletal structure; (11) nuclear regulation of transcription; (12) ribosomal protein synthesis; (13) desmosomal contacts; (14) connexin channel communication; (15) sarcoglycan complex interactions.

5.-As suggested, unnecessary spaces are deleted from the text.

We hope those changes may contribute to improve the final manuscript version.

Reviewer 2 Report

In general, this review by Fernández-Solà is very interesting, since it summarizes current evidence on the effects of alcohol consumption on alcoholic cardiomiopaty. This work is well organized and described. I'm wondering if there is evidence about the different effect of beverage choice. For instance, is there difference between wine, beer or spirits. Moreover, I would suggest to better explain difference between light and heavy drinking.

The paragraph on sex-differences is very interesting.

Beyond previous comments, I would suggest moderate English editing for some confusing sentences and typos.  

Author Response

Response to reviewer  n.2:

We thank this reviewer comments and suggestions.

Point-By-Point response:

1.- Concerning to evidence about the different effect of beverage choice ( wine, beer or spirits) in alcoholic cardomyopathy (ACM).

This aspect was partially included in the previous version (see lines 156 to 158)

“In relation to types of beverage, all may cause ACM, but whine is less damaging compared to other alcohol beverages, probably because his antioxidant polyphenolic content, with molecules such as resveratrol [79-80]”. 

As suggested, we have expanded this subject  in the present version and introduced the following text in section 2.4.  The dose-related effect of ethanol and beverage types on the heart

Concerning the different effects of beverage choice, ACM may develop through consumption of any type of beverage such as wine, beer, or spirits in a lineal dose-dependence relationship with the total lifetime dose of ethanol consumed by an individual [38]. In general, alcoholic patients consuming >90 g of alcohol a day (approximately seven to eight standard drinks per day) for >5 years are at risk for the development of asymptomatic ACM [18]. Wine is considered less damaging compared to other alcohol beverages, probably because of its antioxidant polyphenolic content, with molecules such as resveratrol [79-80]. Consumption of spirits that contain greater ethanol content may easily induce binge drinking and higher cumulated lifetime dose of ethanol, increasing the risk of ACM [19].

2.- Concerning the differences between light and heavy drinking

We have considered this aspect in sections 2.5 and 2.6 as follow:

 2.5. The effects of moderate consumption of ethanol and binge-drinking

      Moderate drinking, considered as consumption of 20-60 g/day in men and 10-40 g/day in women, usually is not associated with significant cardiotoxicity [19, 44]. It does not suppose a risk of ACM development unless consumed in a large period of time (more than 10 years) [19,52]. Moderate alcohol consumption has been associated with lower risk of heart failure in prior studies of healthy individuals [52] and appears equivalent to abstention in improving LV ejection fraction among heavy drinkers with established ACM [81-82].

Binge drinking, defined as the consumption in men of five or more drinks and four or more drinks in women in about two hours, is clearly detrimental for the heart [84]. It brings a person’s blood ethanol concentration (BAC) to 0.08 grams or higher [86].It causes acute myocardial effects with a temporary depression of LV EF evident in experimental [85-86] and clinical models [87-88]. Is more frequent in subjects with LV EF <40% than in those with preserved LV EF [56]. Acute ethanol binge drinking also induces a variety of arrhythmias, known as “Holiday heart Syndrome” [43]. All these acute effects produce impairment on the natural course of chronic ACM [62]. Spirits and other beverages containing a high percentage of alcohol are more detrimental than wine consumption regarding induction of acute cardiac effects [31,80].

2.6. The effect of low-dose ethanol on ACM

Low-dose ethanol consumption, considered as daily consumption of up to 1 standard drink for women and 2 standard drinks for men, has a beneficial effect on preventing coronary heart disease [44], heart failure [87], and global mortality [89] as assessed in multiple clinical and epidemiological studies with a clear “J-shape” curve of effect [58,90]. However, it is possible with low-dose alcohol consumption to achieve an accumulated lifetime dose of ethanol reaching the threshold level to develop ACM in long-term susceptible consumers [19]. This is especially possible in those patients more sensitive to the toxic effects of ethanol on the heart such as women [46] and patients with other systemic diseases related to ethanol (cirrhosis, malnutrition, or neurological damage). In addition, some genetic polymorphisms such as the “DD” isoform for the angiotensin-converting enzyme gene [91] and titin truncated-variants [92] are associated with higher genetic vulnerability to ACM. Therefore, there is no safe dose of ethanol consumption to completely avoid development of alcohol CMP, with complete abstinence being recommended in susceptible subjects [41, 45, 51].

 3.- English language  and stile correction have been performed in the final version of the manuscript.

We hope those changes may contribute to improve the final manuscript version.

Reviewer 3 Report

This paper aims to review the effects of ethanol on alcoholic cardiomyopathy. This review is interesting and could provide valuable information regarding ethanol's effect on other organs besides the gut-liver and neurological systems which are best described. Unfortunately, the spelling, grammar and organization of the article make it difficult to gain the full potential this article could have.  Some suggestions:

-correct the spelling and grammar - for example, authors refer to ethanol as "his" rather than it.

-choose a means to represent ethanol and continue with it throughout the document. The authors interchange ethanol, alcohol,  EtOH, etOH, etc...please be consistent

-more detail regarding ethanol effects on the heart. The information that is provided is very limited and doesn't fully give reader perception of what is known. 

-figure 1 is nice, but it needs a figure legend. The figure is taken directly from the Nature paper, but it has added numbers to it that need an explanation as to what they represent

-section 3.2 has subtitle of autophagy, but there is only 1 sentence regarding autophagy stating that there is insufficient understanding of its role in ACM. Either eliminate autophagy from subtitle or expand upon it.

-figure 2 image is of poor quality. Also it would be nice to have arrows added to figure to point to what is mentioned in the figure legend

-authors refer to total doses of ethanol of 7 or 20 kg/kg body weight. What would this equate to drinks per month?

-treatment of ACM needs more detail regarding diet - there is one sentence regarding Mediterranean diet, but not to how it may be beneficial in the treatment, especially since this diet allows for red wine.

Author Response

Response to reviewer  n.3:

We thank this reviewer comments and suggestions.

Point-By-Point response:

1.- With respect to English corrrection, we agree that many spelling mistakes remain in the previous text version. Sorry, this is because we have performed the English correction in the final version of the manuscript

2.- As suggested, in the present version we chose to represent “ethanol” throughout the document. However, in some cases is difficult to maintain i.e. Alcoholic cardiomyopathy. We explain in Introduction the use of the term “alcohol” as equivalent to “ethanol”.

3.- We have expanded regarding ethanol effects on the heart:.

In section 3 Pathological aspects of ACM

Thus, ethanol enhances permeation in model membranes by interfering with plasma membrane composition and permeability [99], disturbing signaling mechanisms, and activating apoptosis [101], as well as disturbing L-Type Ca2+ channel activity [85-86], Na+/K+ ATPase channel activity [102], Na+/Ca2+ exchanger activity, and Na+ and K+ channel currents [16,29]. Specifically, ethanol disturbs the ryanodine Ca2+ release, the sarcomere Ca2+sensitivity [102-103], the excitation–contraction coupling and myofibrillary structure, and protein expression decreasing heart contraction [85]. Ethanol-induced disruption of ribosomal protein synthesis also contributes to non-contractile protein depletion [104]. Several aspects of mitochondrial function, including respiratory complex activities and mitochondrial-dependent oxidative damage and apoptosis are also induced by ethanol [26,100]. Myocyte cytoskeletal structure [21], connexin channel communication, and desmosomal contacts are affected by ethanol, causing structural cell instability [105]. Ethanol may induce changes in nuclear regulation of transcription with a dose-dependent translocation of NFkB into the nucleus [106]. The resulting effect in those multiple sites may be additive and synergistic, increasing the final damage [20, 52] (Figure 1).

The resulting effect in those multiple sites may be additive and synergistic, increasing the final damage [20, 52].Figure 1  

See also apoptosis and autophagy section (Section 3.2)

 4.- We have incorporated a figure legend to Figure 1, explaining what numbers represent.

Fig 1 Legend: Cardiac myocytes are excitable cells with complex signaling and contractile structures and are highly sensitive to the toxic effect of alcohol on: (1) plasma membrane composition and permeability, signaling, and activation of apoptosis; (2) L-Type Ca2+ channel activity; (3) Na+/K+ ATPase channel activity; (4) Na+/Ca2+ exchanger activity; (5) Na+ channel currents; (6) K+ channel currents; (7) ryanodine Ca2+ release; (8) sarcomere Ca2+ sensitivity, excitation–contraction coupling, myofibrillary structure, and protein expression; (9) several aspects of mitochondrial function, including respiratory complex activities; (10) cytoskeletal structure; (11) nuclear regulation of transcription; (12) ribosomal protein synthesis; (13) desmosomal contacts; (14) connexin channel communication; (15) sarcoglycan complex interactions.

5.-We have expanded the autophagy effect of ethanol in ACM in section 3.2.

Recently, apoptosis and necrosis have been also attributed to autophagy in ACM [18]. In order to maintain cardiac homeostasis, removal of defective organelles and cell debris by autophagy is essential both in physiological and pathological conditions [114]. Dysregulated excessive autophagy together with other factors such as oxidative stress, neurohormonal activation, and altered fatty acid metabolism contributes to cardiac structural and functional damage following alcoholism. This influences the maintenance of cardiac geometry and contractile function, increasing the development of ACM [120]. In ACM, protein degradation with sarcomere disarray and contractile protein loss has been suggested to be a key point of autophagy induction [18]. Different pathogenic hypotheses have been suggested such as the pivotal role of acetaldehyde [121], the role of oxidative stress and stress signaling cascades [109], and translocation of NFkB into the nucleus [106]. Although the mechanism of action behind autophagy and its signaling regulatory cascades remains elusive in ACM [120], its understanding may contribute to better identifying molecular mechanisms underlying early stages of alcoholic cardiomyopathy and suggest novel strategies to counteract integrated risk of cardiotoxicity in chronic alcohol consumption [106].

And also in the Section 5, ACM Treatment, at the end, Lines 502-507

“Pharmacological restoration of autophagic reflux by inhibition of soluble epoxide hydrolase has been described to ameliorate chronic-ethanol-induced cardiac fibrosis in an in vivo swine model [151]. In addition to these, stem-cell therapy tries to improve myocyte regeneration [146,152]. However, these new strategies have not yet demonstrated their real effectiveness in clinical trials, require further evaluation, and are not still approved for clinical use [147].”

6.- We add arrows to Figure 2. In our review version we appreciate good resolution in the images of this figure. This poor quality that you report may be due to a copy default ?.

7.- The equivalent of 7 to 20 Kg/Kg body weight of ethanol consumption  is 60 to 180 drinks per month.

This has been reflected in the text:

Section 2.4 Line n. 61

Section 3.7 Line n. 372

8.- As suggested, we have increased the dietary aspects in treatment of ACM, considering the conflict of red wine in Mediterranean diet.

See section #5,  lines 418 to 434.

A Mediterranean diet, based on monounsaturated fats from olive oil, fruits, vegetables, whole grains, and legumes/nuts has demonstrated to be beneficial for primary prevention of global cardiovascular events (myocardial infarction, stroke, or death from cardiovascular causes) [80,140-141]. However, since it includes moderate alcohol consumption of red wine, this aspect should be clearly avoided in subjects affected by ACM. The exact mechanism by which an increased adherence to the traditional Mediterranean diet exerts its favorable effects is not known. However, its beneficial cardiovascular effect may be caused by different factors including lipid-lowering, protection against oxidative stress, inflammation and platelet aggregation, modification of hormones and growth factors, inhibition of nutrient sensing pathways by specific amino acid restriction, and gut microbiota-mediated production of metabolites influencing metabolic health [142].

       In ACM it is relevant to consider treatment of the other alcohol-induced systemic damage, such as liver cirrhosis, malnutrition, and vitamin and electrolyte disturbances [4,11,52]. Notably, in patients with a history of chronic alcohol consumption complicated by significant myocardial dysfunction and chronic malnutrition, re-feeding syndrome, may increase the cardiac dysfunction. Therefore, physicians should be aware of the risk of new cardiomyopathy in patients with these overlapping diagnoses [143]

We hope those changes may contribute to improve the final manuscript version.

Round 2

Reviewer 1 Report

The manuscript is well revised.

Author Response

We thank this reviewer for nice considertions

Reviewer 3 Report

Thank you for revising your manuscript based on reviewer comments.

While there is great improvement, there are still some items that should be considered:

-representation of statements for alcohol consumption in gm/d or gm/kg into drinks/day were only changed for some of the statements. there are still others that would benefit from clarification (e.g., section 2.4 (5 kg ethanol is this amount correct? per kg body weight for women; section 2.5, section 4, etc)

-editing for English language has improved the document, but there are still corrections that need to be made

Author Response

We thank this reviewer for those new comments

POINT-BY-POINT RESPONSE:

  • We have established in the article an equivalence between g/day to drinks /day in order to not repeat it along the article                                             See lines 161 to 165:
  • "Concerning the different effects of beverage choice, ACM may develop through consumption of any type of beverage such as wine, beer, or spirits in a lineal dose-dependence relationship with the total lifetime dose of ethanol consumed by an individual [38]. In general, alcoholic patients consuming >90 g of alcohol a day (approximately seven to eight standard drinks per day, considering an standad drint 12-15 g of alcohol) for >5 years are at risk for the development of asymptomatic ACM [18]."                     
  • With reference to cummulated dose of alcohol (Kg alcohol per Kg of body weight), the equivalence is not direct, since it depends on the individual weight and the time of consumption. As suggested, we have changes and amplified these statemets in section 2.4 lines 54 to 57:

        "This is a longstanding accumulated effect that usually appears when a                 subject has in their lifetime consumed more than 7 Kg of ethanol per Kg            of body weight in men (equivalent to 60 drinks per month), and 5 Kg of              ethanol per Kg of body weight in women (equivalent to 43 drinks per                  month) [19, 46]".

  • and section 2.5 (lines 172 to 174):                                                          "Moderate drinking, considered as consumption of 20-60 g/day in men (1,5 to 4 standard drinks) and 10-40 g/day in women (1/2 to 3 standard drinks), usually is not associated with significant cardiotoxicity [19, 44]."            
  • and section 3.7:  "This is usually after more than 20 years of high ethanol consumption at cumulated lifetime doses higher than 20 Kg ethanol/Kg body weight, equivalent to 180 drinks per month [52,133]."                            
  • and section 4:                                                                                         (line 374 to 376) "On the contrary, subjects who continue drinking at moderate to high doses (more than 60 g ethanol/day in men -equivalent to 4 estandard drinks-, and 40 g of ethanol/day in women - equivalent to 2,5 standard drinks- ..."                                                                                                                     
  • The cummulated ammount of 5 Kg/ethanol per Kg/body weight is correct in susceptible women, considering a consumption of 40 g/dayfor 20 years in a women with 55 Kg of body weight, This is the lowest thershold that we have observed in suceptble women who develop ACM.